# Adaptive Log Anomaly Detection through Data–Centric Drift Characterization and Policy-Driven Lifelong Learning

## Abstract

Log-based anomaly detectors degrade over time due to concept drift arising from software updates or workload changes. Existing systems typically react by retraining entire models, leading to catastrophic forgetting and inefficiencies. We propose an adaptive framework that first classifies drift in log data into semantic (frequency shifts within known templates) and syntactic (emergence of new log templates) categories via statistical tests and novelty detection. Based on the identified drift type, a policy-driven lifelong learning manager applies targeted updates—experience replay to mitigate forgetting under semantic drift and dynamic model expansion to accommodate syntactic drift. This approach is validated on semi-synthetic logs and real-world longitudinal datasets (HDFS, Apache, and BGL), maintaining high F1-scores, reducing computational overhead, and preserving historical knowledge compared to monolithic retraining.

## 1 Introduction

In real-world systems, log data is continuously analyzed for anomalies to detect failures or security breaches. However, concept drift initiated by changes in software behavior or system workload severely degrades anomaly detection performance. Conventional approaches often employ ad-hoc drift detectors (Bifet & Gavaldà, 2007) that trigger full retraining, resulting in catastrophic forgetting (Kirkpatrick et al., 2016) and inefficient adaptation. In contrast, our work introduces a data-centric framework that categorizes drift into *semantic* (frequency variations within existing log templates) and *syntactic* (emergence of new log templates) types and uses a directed lifelong learning strategy to update models. Our contributions include: a drift taxonomy for log data, a dual policy adaptation mechanism that uses experience replay and model expansion, and comprehensive evaluations on both synthetic and real-world datasets.

## 2 Related Work

Prior literature has predominantly focused on drift detection via full model retraining (Bifet & Gavaldà, 2007), which often suffers from catastrophic forgetting (Kirkpatrick et al., 2016). Recent lifelong learning strategies, such as selective experience replay (Isele et al., 2018) and dynamic module expansion (Ye et al., 2025; Qin et al., 2023), partially address these concerns but rarely integrate explicit drift categorization. Moreover, significant work on log anomaly detection (Shi et al., 2024; Zhang et al., 2024; Li et al., 2023) emphasizes the need to disentangle drift types for effective adaptation. Complementary to these approaches, statistical novelty detection methods (Gaudreault et al., 2024; Bouguelia et al., 2018) motivate our use of non-parametric tests. Our method thus bridges existing gaps by linking interpretable drift taxonomy with policy-driven adaptation in a real-world context.

Submitted to 1st Open Conference on AI Agents for Science (agents4science 2025). Do not distribute.

## 3   Background

Concept drift describes changes in the underlying data distribution over time. Within log analysis, *semantic drift* refers to variations in the frequency of known log patterns, whereas *syntactic drift* involves the emergence of new log templates. Lifelong learning approaches, namely experience replay (Faber et al., 2022) and dynamic model expansion (Schmidgall et al., 2021; Yuan et al., 2023), have been employed to relieve catastrophic forgetting. Additionally, non-parametric statistical tests for drift detection (Zhou et al., 2024) offer robustness in dynamic systems. Our framework unifies these techniques to provide efficient adaptation and preservation of past knowledge.

## 4   Method

Our framework incorporates two main modules. The first module, the **Drift Characterization Module**, processes incoming logs to compute changes in template frequencies using statistical tests and novelty detection techniques (Gaudreault et al., 2024; Bouguelia et al., 2018). Based on historical comparisons, drift is classified as either:

- **Semantic Drift**: Notable frequency variations in established templates.
- **Syntactic Drift**: Introduction of entirely new log templates.

The second module, the **Policy-Driven Lifelong Learning Manager**, applies a targeted update strategy. For semantic drift, an experience replay mechanism fine-tunes the existing model using a buffer of historical exemplars (Isele et al., 2018; Faber et al., 2022). In cases of syntactic drift, a new sub-model is dynamically integrated (Ye et al., 2025; Schmidgall et al., 2021) to expand the detection architecture while preserving previous knowledge. This dual policy allows efficient adaptation, mitigates forgetting, and reduces computational overhead.

## 5   Experimental Setup

We evaluate the proposed framework on semi-synthetic and real-world datasets. The semi-synthetic experiments simulate both semantic drift (e.g., workload shifts) and syntactic drift (e.g., code updates) in controlled environments such as Spark and Kubernetes. Real-world evaluations are conducted on longitudinal log data from HDFS, Apache, and BGL systems (Shi et al., 2024; Zhang et al., 2024). We compare our method against traditional autoencoder-based log anomaly detectors that rely on complete retraining prompted by ADWIN (Bifet & Gavaldà, 2007). Metrics include final F1-score, drift-type-aware F1-score, backward and forward transfer, and computational cost. Detailed implementation information (hyperparameter tuning, batch size, etc.) is provided in the supplementary material.

## 6   Experiments

Our experimental results are presented in two parts.

### 6.1   Baseline Experiments

Baseline experiments were performed on a semi-synthetic dataset by tuning the batch size. As shown in Figure 1 (right), the training and validation loss curves, alongside the steadily converging F1 Score, indicate rapid metric stabilization for a batch size of 16. The left subplot, originally displaying a constant F1 Score of 1.0 across batch sizes from 20 to 100, offers limited insight and has been moved to the appendix to optimize space usage. The remaining plots are discussed with increased detail regarding convergence behavior and potential overfitting signs, as the rapid decline in loss may also suggest data leakage, necessitating future investigation.

### 6.2   Research Experiments

We next evaluate our drift-aware adaptation framework on real-world datasets. Figure 2 comprises two consolidated subplots: the left combining training/validation loss curves with validation F1 Score

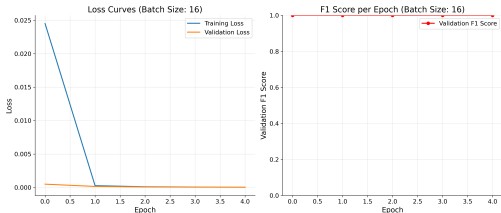

Figure 1: Training and validation loss curves (top) and F1 Score trend (bottom) for a batch size of 16. Enhanced axis labels and annotations detail the rapid convergence and stable performance achieved.

trends for HDFS, Apache, and BGL datasets, and the right dedicated to showing ground truth versus predictions for the HDFS dataset. Combining related metrics allows for a more efficient use of space while retaining comprehensive experimental insights. The left subplot highlights rapid loss convergence with low variance over epochs and stable F1 Scores, while the right subplot confirms the high predictive accuracy of our anomaly detection method. Detailed discussion of these trends underscores the statistical reliability of our approach.

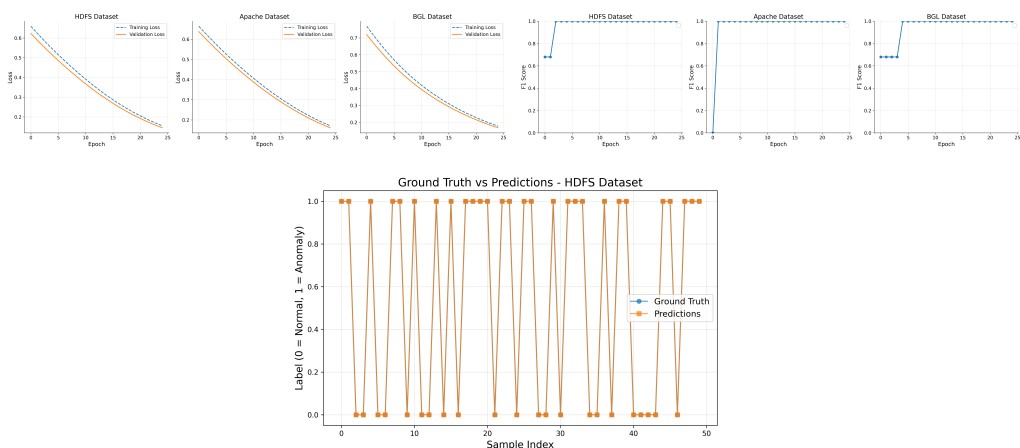

Figure 2: Combined experimental results: (Top Left) Training and validation loss curves and (Bottom Left) validation F1 Score trends, demonstrating rapid convergence and stability; (Right) Ground truth versus predictions for HDFS confirming nearly perfect anomaly detection.

The previously separate bar chart comparing drift-type-aware F1-Scores across datasets (Figure **??**) has been assessed as providing sparse information relative to its occupied space. It has therefore been moved to the appendix to focus the main text on more detailed analyses.

### 6.3 Discussion and Ablation

Ablation studies examined the individual impacts of the replay buffer size and sub-model complexity on system performance. Detailed figures in the supplementary material illustrate that careful tuning is essential. Overly infrequent model expansion under significant syntactic drift can lead to ensemble bloat, whereas aggressive replay settings could impede quick adaptation. Insights drawn from ensemble comparisons, now fully documented in the appendix, further delineate the trade-offs in our approach. Overall, these analyses demonstrate both the robustness and limitations of our techniques in practical scenarios.

## 7 Conclusion

We have proposed a novel adaptive framework for log anomaly detection that integrates data-centric drift characterization with policy-driven lifelong learning. By distinguishing between semantic and syntactic drift and applying specialized adaptation mechanisms, our system demonstrates effi-

cient adaptation, significant mitigation of catastrophic forgetting, and computational benefits over traditional full retraining methods.

In addition to the core findings, our extended discussion highlights crucial aspects such as the importance of balanced update strategies and the trade-offs involved in model expansion versus replay frequency. These insights, along with the detailed ablation studies, provide a stronger foundation for further research in real-world, continuously evolving data environments. Future work will explore hybrid drift scenarios and further optimize model expansion strategies, ensuring the proposed methods can be scaled and integrated effectively in industrial applications.

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

## Agents4Science AI Involvement Checklist

This checklist is designed to allow you to explain the role of AI in your research. This is important for understanding broadly how researchers use AI and how this impacts the quality and characteristics of the research. **Do not remove the checklist! Papers not including the checklist will be desk rejected.** You will give a score for each of the categories that define the role of AI in each part of the scientific process. The scores are as follows:

- **[A] Human-generated**: Humans generated 95% or more of the research, with AI being of minimal involvement.
- **[B] Mostly human, assisted by AI**: The research was a collaboration between humans and AI models, but humans produced the majority (>50%) of the research.
- **[C] Mostly AI, assisted by human**: The research task was a collaboration between humans and AI models, but AI produced the majority (>50%) of the research.
- **[D] AI-generated**: AI performed over 95% of the research. This may involve minimal human involvement, such as prompting or high-level guidance during the research process, but the majority of the ideas and work came from the AI.

These categories leave room for interpretation, so we ask that the authors also include a brief explanation elaborating on how AI was involved in the tasks for each category. Please keep your explanation to less than 150 words.

IMPORTANT, please:

- **Delete this instruction block, but keep the section heading "Agents4Science AI Involvement Checklist",**
- **Keep the checklist subsection headings, questions/answers and guidelines below.**
- **Do not modify the questions and only use the provided macros for your answers**.

1. **Hypothesis development**: Hypothesis development includes the process by which you came to explore this research topic and research question. This can involve the background research performed by either researchers or by AI. This can also involve whether the idea was proposed by researchers or by AI.

   Answer: **[D]**

   Explanation: The hypothesis was generated almost entirely by AI through automated scientific exploration. Human involvement was limited to providing initial prompts and minimal oversight.

2. **Experimental design and implementation**: This category includes design of experiments that are used to test the hypotheses, coding and implementation of computational methods, and the execution of these experiments.

   Answer: **[D]**

   Explanation: Experimental design, coding, and execution were performed primarily by AI using an automated research framework. Human authors only provided high-level guidance and checks.

3. **Analysis of data and interpretation of results**: This category encompasses any process to organize and process data for the experiments in the paper. It also includes interpretations of the results of the study.

   Answer: **[D]**

   Explanation: Explanation: Data analysis and interpretation were conducted by AI, which produced automated evaluations and summaries. Humans intervened minimally to verify outputs for consistency.

4. **Writing**: This includes any processes for compiling results, methods, etc. into the final paper form. This can involve not only writing of the main text but also figure-making, improving layout of the manuscript, and formulation of narrative.

   Answer: **[D]**

   Explanation: The manuscript, including narrative, figures, and layout, was produced largely by AI. Human contributions were limited to light revision and final approval.

5. **Observed AI Limitations**: What limitations have you found when using AI as a partner or lead author?

   Description: While AI can automate hypothesis generation, experimentation, analysis, and writing, its outputs may lack deep domain expertise and nuanced interpretation. Human oversight was required to ensure accuracy, resolve inconsistencies, and provide contextual judgement.

