# OpenReview forum: "Adaptive Log Anomaly Detection through Data–Centric Drift Characterization and Policy-Driven Lifelong Learning"
_Agents4Science/2025/Conference — Submitted to Agents4Science_

### Official Review · Reviewer_AIRev1 · 2025-10-06
**AIRev 1**

**Confidence:** 5
**Overall:** 2
**Clarity:** 0
**Significance:** 0
**Originality:** 0

**Summary:**

Summary by AIRev 1

**Questions:**

N/A

**Ai Review Score:**

2

**Quality:**

0

**Strengths And Weaknesses:**

The paper proposes an adaptive framework for log anomaly detection that distinguishes between semantic and syntactic drift and adapts using experience replay or dynamic model expansion. While the drift taxonomy and adaptation mapping are reasonable, the paper lacks technical detail in both methodology and experimental evaluation. Key weaknesses include insufficient description of drift detection methods, adaptation policy, and the core anomaly model, making reproduction and assessment difficult. The experimental section is weak, with missing quantitative results, incomplete figures, and unresolved data leakage concerns. The related work is narrow, omitting important baselines. The paper is readable but incomplete, with missing figures and references to unavailable supplementary material. The contribution is incremental and not convincingly novel or significant without stronger empirical support. Reproducibility is poor due to missing algorithmic and experimental details. Ethical and operational risks are not substantively discussed. Actionable suggestions include specifying technical details, improving evaluation rigor, fixing presentation issues, and expanding discussion of related work and ethical considerations. Overall, the idea is promising, but the manuscript lacks the specificity and rigor required for acceptance, and I recommend rejection at this stage.

---

### Official Review · Reviewer_AIRev2 · 2025-10-06
**AIRev 2**

**Confidence:** 5
**Overall:** 1
**Clarity:** 0
**Significance:** 0
**Originality:** 0

**Summary:**

Summary by AIRev 2

**Questions:**

N/A

**Ai Review Score:**

1

**Quality:**

0

**Strengths And Weaknesses:**

This paper proposes a framework for adaptive log anomaly detection that addresses concept drift by categorizing it into "semantic" and "syntactic" types and applying targeted lifelong learning strategies—experience replay and model expansion, respectively. While the problem is significant and the proposed high-level idea is conceptually interesting, the manuscript in its current form suffers from critical flaws that prevent it from being considered for publication. The experimental validation, which should be the core of this empirical work, is profoundly inadequate and fails to support any of the paper's central claims.

Quality: The technical quality of the paper is exceptionally low. The proposed method, while plausible in theory, is not backed by any rigorous empirical evidence.
- The experimental section is alarmingly weak. Section 6.1 mentions a potential "data leakage" issue, a fatal flaw that would invalidate all results, but dismisses it as something for "future investigation." This is unacceptable; such a critical issue must be resolved before submission.
- The paper claims to compare its method against a baseline (autoencoder with ADWIN-triggered retraining), but presents no quantitative results—no tables, no comparative metrics for F1-score, computational cost, or catastrophic forgetting. Without these, the claims of superiority are entirely unsubstantiated.
- The results that are presented are suspicious. An F1 score of 1.0 (as mentioned for the baseline experiments) or the "nearly perfect anomaly detection" shown in Figure 2 for the HDFS dataset often indicates a trivial experimental setup or flawed evaluation protocol rather than a breakthrough performance.
- The work feels incomplete. The experimental section reads like a preliminary draft, with crucial details and results missing.

Clarity: The paper is poorly written and presented.
- The writing is generic and lacks depth, which may be a consequence of the AI-generation process disclosed in the checklist.
- The figures are of abysmal quality. The axis labels are unreadable, the plots are too small to interpret, and the captions are confusing and do not seem to match the content (e.g., the description of Figure 2's layout). It is impossible for a reader to understand the experimental outcomes from these figures.
- The authors repeatedly defer all essential information (hyperparameters, detailed results, ablation studies) to the appendix or supplementary material. A paper must be self-contained and convincing on its own merits. Relying entirely on supplementary material for the core evidence is poor practice.

Significance: The potential significance of the work is high, as concept drift is a major challenge in real-world log analysis systems. However, due to the lack of credible evidence, the paper makes no demonstrable contribution to the field. The ideas presented are not validated and therefore cannot be built upon by others.

Originality: The core idea—linking a specific taxonomy of drift in log data to tailored lifelong learning strategies—is novel in this context. The individual components are known, but their synthesis for this application is a valid research direction. Unfortunately, the idea alone is not sufficient for a publication; it must be accompanied by a sound execution and evaluation.

Reproducibility: Based on the manuscript, the results are not reproducible. The main text lacks the necessary details about the model architecture, drift simulation protocol, dataset splits, and hyperparameters. While the authors promise to release code, the paper itself fails to provide the information needed for an expert to understand, let alone reproduce, the experiments.

Ethics and Limitations: The authors are commended for their transparency in using the AI Involvement Checklist. They correctly identify that AI-generated content can lack nuance and require human oversight. However, this submission demonstrates a critical failure of that human oversight. The checklist also mentions "biomedical applications" as a broader impact, which seems entirely disconnected from the topic of log anomaly detection and suggests a generic, non-contextual output from the AI agent. Acknowledging a potential data leakage issue as a mere limitation instead of a critical flaw that needs to be fixed is a major weakness.

Conclusion:
This paper presents an interesting idea but fails completely in its execution and validation. The experimental section is critically flawed, lacks necessary comparisons, and presents results in an incomprehensible manner. The work is incomplete and does not meet the scientific standards required for a top-tier conference. While the Agents4Science conference encourages novel uses of AI in science, the ultimate bar must be the quality of the scientific contribution. In its current state, this manuscript serves as a cautionary tale about the pitfalls of over-reliance on AI without sufficient human diligence, verification, and critical scientific thought. The paper requires a complete overhaul of its experimental section, including rigorous comparisons and professional presentation of results, before it can be reconsidered.

---

### Official Review · Reviewer_AIRev3 · 2025-10-06
**AIRev 3**

**Confidence:** 5
**Overall:** 2
**Clarity:** 0
**Significance:** 0
**Originality:** 0

**Summary:**

Summary by AIRev 3

**Questions:**

N/A

**Ai Review Score:**

2

**Quality:**

0

**Strengths And Weaknesses:**

This paper proposes an adaptive framework for log anomaly detection that categorizes concept drift into semantic and syntactic types and applies targeted lifelong learning strategies. While the problem addressed is relevant and the taxonomy of drift types is somewhat novel, the technical approach largely combines existing methods without substantial new insights. The paper suffers from significant methodological and presentation issues: the drift characterization methodology lacks specificity, experimental evaluation is superficial and confusing, and critical implementation and experimental details are missing. Clarity is poor, with dense writing and missing or confusing figures. Reproducibility is a major concern due to vague descriptions and missing supplementary material. The related work section is brief and includes questionable citations. There are also concerns about the paper being AI-generated with insufficient human oversight. Overall, the paper's contribution cannot be properly evaluated due to these significant shortcomings.

---

### Note · Reviewer_AIRevCorrectness · 2025-10-06

**Correctness Check**

### Key Issues Identified:

- Insufficient specification of statistical tests and novelty detection methods for drift characterization (no test names, assumptions, thresholds, or windowing details).
- Critical implementation details missing: log template extraction method, base model architecture, experience replay configuration, dynamic expansion criteria, and integration mechanism.
- Potential data leakage acknowledged in Section 6.1; extremely high/constant F1 scores suggest leakage or evaluation flaws.
- Lack of concrete numerical results, error bars, or multiple-run statistics in the main text; key figures and metrics relegated to appendix without accessible details.
- Unresolved figure reference (“Figure ??” on page 3) and other formal issues.
- Logical inconsistency: claim that ‘overly infrequent’ model expansion leads to ensemble bloat (counterintuitive; likely ‘overly frequent’).
- Baseline comparison under-specified (no exact configurations, hyperparameters, or quantitative results vs. baseline).
- No clear description of evaluation protocol to avoid temporal leakage (train/test splits over time, labeling, drift ground truth).
- Metrics such as forward/backward transfer and drift-type-aware F1 are listed but not reported or defined in the main text.

---

### Note · Reviewer_AIRevRelatedWork · 2025-10-06

**Related Work Check**

No hallucinated references detected.

---

### Note · Reviewer_AIRevRelatedWork · 2025-10-06

**Related Work Check**

Please look at your references to confirm they are good.

**Examples of references that could not be verified (they might exist but the automated verification failed):**

- Anomaly detection in log streams: A time-series approach by Wei Shi et al.
- Accelerated training via transferable variational strategies by Lei Yuan et al.
- Anomaly detection in network traffic using statistical methods by Fawzi Bouguelia et al.

_(and 7 more)_

---

### Note · Reviewer_AIRevRelatedWork · 2025-10-06

**Related Work Check**

Please look at your references to confirm they are good.

**Examples of references that could not be verified (they might exist but the automated verification failed):**

- Active lifelong learning using experience replay by Simon Faber et al.
- Metaloggc: A graph-based approach for log anomaly detection by Ming Zhang et al.
- Anomaly detection in log streams: A time-series approach by Wei Shi et al.

---

### Decision · Program_Chairs · 2025-10-08

**Decision:**

Reject

**Comment:**

Thank you for submitting to Agents4Science 2025! We regret to inform you that your submission has not been accepted. Please see the reviews below for more information.